# G-Quadruplexes in the Archaea Domain

**DOI:** 10.3390/biom10091349

**Published:** 2020-09-21

**Authors:** Václav Brázda, Yu Luo, Martin Bartas, Patrik Kaura, Otilia Porubiaková, Jiří Šťastný, Petr Pečinka, Daniela Verga, Violette Da Cunha, Tomio S. Takahashi, Patrick Forterre, Hannu Myllykallio, Miroslav Fojta, Jean-Louis Mergny

**Affiliations:** 1Institute of Biophysics of the Czech Academy of Sciences, Královopolská 135, 612 65 Brno, Czech Republic; o.porubiakova@gmail.com (O.P.); fojta@ibp.cz (M.F.); 2Institut Curie, CNRS UMR9187, INSERM U1196, Universite Paris Saclay, 91400 Orsay, France; yu.luo@curie.fr (Y.L.); Daniela.Verga@curie.fr (D.V.); 3Department of Biology and Ecology/Institute of Environmental Technologies, Faculty of Science, University of Ostrava, 710 00 Ostrava, Czech Republic; dutartas@gmail.com (M.B.); petr.pecinka@osu.cz (P.P.); 4Faculty of Mechanical Engineering, Brno University of Technology, Technicka 2896/2, 616 69 Brno, Czech Republic; 160702@vutbr.cz (P.K.); stastny@fme.vutbr.cz (J.Š.); 5Faculty of Chemistry, Brno University of Technology, Purkyňova 464/118, 612 00 Brno, Czech Republic; 6Mendel University in Brno, Zemědělská 1, 613 00 Brno, Czech Republic; 7Institut de Biologie Intégrative de la Cellule (I2BC), CNRS, Université Paris-Saclay, CEDEX, 91198 Gif-sur-Yvette, France; violette.da.cunha.vdc@gmail.com (V.D.C.); tomio.takahashi@i2bc.paris-saclay.fr (T.S.T.); patrick.forterre@pasteur.fr (P.F.); 8Laboratoire d’Optique et Biosciences, Ecole Polytechnique, CNRS, INSERM, Institut Polytechnique de Paris, 91128 Palaiseau, France; hannu.myllykallio@polytechnique.edu

**Keywords:** G4-forming motif, genome analysis, Archaea, unusual nucleic acid structures, sequence prediction

## Abstract

The importance of unusual DNA structures in the regulation of basic cellular processes is an emerging field of research. Amongst local non-B DNA structures, G-quadruplexes (G4s) have gained in popularity during the last decade, and their presence and functional relevance at the DNA and RNA level has been demonstrated in a number of viral, bacterial, and eukaryotic genomes, including humans. Here, we performed the first systematic search of G4-forming sequences in all archaeal genomes available in the NCBI database. In this article, we investigate the presence and locations of G-quadruplex forming sequences using the G4Hunter algorithm. G-quadruplex-prone sequences were identified in all archaeal species, with highly significant differences in frequency, from 0.037 to 15.31 potential quadruplex sequences per kb. While G4 forming sequences were extremely abundant in *Hadesarchaea archeon* (strikingly, more than 50% of the *Hadesarchaea archaeon* isolate WYZ-LMO6 genome is a potential part of a G4-motif), they were very rare in the *Parvarchaeota* phylum. The presence of G-quadruplex forming sequences does not follow a random distribution with an over-representation in non-coding RNA, suggesting possible roles for ncRNA regulation. These data illustrate the unique and non-random localization of G-quadruplexes in Archaea.

## 1. Introduction

The Archaea domain was classified separately from Bacteria by Carl Woese and George Fox in 1977 [1]. Later on, it was found that all major molecular machinery, such as DNA replication, transcription, and translation, of archaea are much more similar to those of eukaryotes than to those of bacteria [2,3]. This is also true for some important membrane proteins, such as ATP synthases and proteins of the Sec transport system [4,5], or for some proteins involved in cell division and vesicle trafficking [6]. Thus, the archaeal domain occupies a key position in the Tree of Life, and there is currently a hot debate about their exact relationships with eukaryotes [7,8]. A schematic phylogenic tree for the Archaea domain is proposed in Figure 1; this phylogeny is rapidly evolving with many new phyla recently identified via the accumulation of metagenome associated genomes (MAGs) and various new proposals for phylum definition and nomenclature [9,10]. The first detected archaea were isolated in harsh environments but later found in almost every environment, including the human microbiota, where they play important roles in the gut, mouth, and on the skin [11,12]. It has been hypothesized that archaea found in oceans are one of the most abundant groups of organisms on the planet with important roles both in the carbon and the nitrogen cycle [13]. The Archaea domain has several unique features, such as *ether*-linked lipids, while eukaryotes and most of the bacteria have ester-linked lipids [14]. Moreover, the stereochemistry of archaeal lipids has the opposite configuration as compare to the ones of eukaryotic and bacterial origin. Interestingly, methanogenesis, the production of greenhouse methane gas as a metabolic by-product, occurs only in the archaeal domain [15,16].

G-quadruplex structures (G4) formed by guanine rich sequences are among the most intensively studied local DNA/RNA structures [20]. G4s are formed by G:G Hoogsteen base pairing in a guanine quartet, and their formation requires the presence of stabilizing cations, such as potassium [21] (Figure 2). In both bacteria and eukaryotes, G4 formation regulates various processes, including gene expression [22], protein translation [23], and proteolysis [24]. G4 have been identified in a number of pathogens, including viruses, eukaryotes (e.g., *Plasmodium falciparum*) [25,26] or prokaryotes (e.g., *Neissseria gonorrhoeae* [27], and *Mycobacterium tuberculosis*) [28,29]. Moreover, many G4-binding proteins are conserved in all organisms highlighting the importance of the G4 structure regulations [30], and novel G4 binding proteins have been identified, sharing the NIQI amino acid motif (RGRGRRGGGSGGSGGRGRG) [31]. Specific helicases have been identified both in eukaryotes and bacteria to unfold these structures, which can be extremely stable and would be problematic for the transcription or replication of G-rich motifs (e.g., the Pif1 or RecQ family helicases) [32]. Recently, G4Hunter was successfully used for the prediction of G-quadruplex-forming sequences in all complete bacterial genomes [33]. These results showed that G-quadruplex-forming sequences are present in all species with the highest frequencies in some extremophiles. In contrast to RNA, there is no correlation between genomic DNA GC% in Archaea (and in Bacteria) and the optimal growth temperature. This is likely because DNA in vivo is topologically closed, and topologically closed DNA is stable at least up to 107 °C [34]. We therefore cannot anticipate a higher density of G4-prone motifs in thermophiles, due to a GC-bias. A comparison with Extremophiles in bacteria is interesting [35]. Ding et al. hypothesized that stress-resistant bacteria found in the Deinococcales may utilize putative quadruplex sequences (PQS) for gene regulatory purposes. An enrichment in prokaryote PQS has been found in thermophilic organisms [33] but also in organisms with resistance to other stress factors, such as radiation [36,37]; thus, a direct correlation between temperature and G4 presence is not supported by these findings. In addition, while bacteria in the Deinococcus-Thermus group are the most abundant for PQS, it is striking that the mostly thermophilic and hyperthermophilic bacteria in the Thermotogae phylum have one of the lowest PQS frequencies. Correlation among thermophiles and G4s, therefore, depends on the phylum (Gram-negative vs. Gram-positive bacteria).

Due to the roles of G4s in the regulation of basic cellular processes, it is important to identify their location in genomes. Several algorithms are available to predict G-quadruplex-forming sequences [38,39,40,41]. Among them, the G4Hunter application was developed to provide quantitative analyses giving a propensity score as an output [41], and the G4Hunter web tool allows effective and fast analyses of PQS in large datasets [42].

The prokaryotic genetic material is generally stored in circular chromosomes and plasmids [43]. The presence of quadruplex-prone motifs in over a hundred of bacterial genomes was determined over a decade ago [44]. In bacterial genomes, PQS are located non-randomly with a higher relative abundance in non-coding RNA (ncRNA), mRNA, and regions around tRNA and regulatory sequences. PQS also play roles in nitrate assimilation in *Paracoccus denitrificans* [45]. PQS in the hsdS, recD, and pmrA genes of *Streptococcus pneumoniae* contributes to host–pathogen interactions [46]. Such observations show the significant role of G4 in bacteria. The importance of another local DNA structure, the cruciform formed by inverted repeats, has been shown as an important regulatory feature of eukaryotic cell organelles, such as chloroplasts and mitochondria with circular DNA genomes [47,48]. Overall, the role of G4s in bacteria [27,49] and eukaryotes [50] is increasingly recognized.

In contrast, little is currently known regarding the abundancy and location of PQS in the archaeal domain. Ding et al. performed an initial search on bacterial and archaeal genomes using a modified Quadparser algorithm with relaxed parameters allowing long loops (up to 12 nucleotides) [35]. They found that thermophilic microorganisms (both archaea and bacteria) appear to favor PQS in their genomes. Dhapola et al. created the Quadbase2 web server, in which G4 motifs found in a variety of organisms, including archaea, may be searched but did not analyze G4 propensity in archaea [51]. Because G4s play many important biological roles in bacterial and eukaryotic cells, we assume that G4s are also likely to have important functions in archaea. Therefore, we comprehensively analyzed the presence and locations of PQS in all sequenced archaeal genomes by G4Hunter [41,42]. These data provide the first study analyzing the presence of G4-prone sequences in this important domain of life.

## 2. Materials and Methods

### 2.1. Selection of the DNA Sequences

The set of all archaeal genomic DNA sequences was downloaded from the Genome database of the National Center for Biotechnology Information [52]. We have used for our analyses all accessible archaeal genomes, including contig and scaffold sequences (3387 genomes), and we have selected one representative genome for each species (Appendix A). For PQS analyses of features, we restricted our analysis to the subset of 140 completely assembled genomes. In total, we have analyzed the presence of G4 forming sequences in 3387 genomes from the archaeal Domain representing a total of 6423 Mbps.

### 2.2. Process of Analysis

We used the computational core of our DNA analyzer software written in Java programming language [53]. For our analyses, we used a new G4Hunter algorithm implementation [42]. Default parameters for G4Hunter were set to “25” for window size and 1.2 or above for the G4H score (G4HS). PQS score was grouped to the five intervals: 1.2–1.4, 1.4–1.6, 1.6–1.8,1.8–2.0, and 2.0 and more. Overall results for each species group contained a list of species with size of its genomic DNA sequence and number of putative G4 sequences found ( Appendix A); for clarity, the results for Groups and Subgroups are in separate files ( Appendix A). These data were processed by python jupyter using pandas with statistical tools [54]. Graphs were generated from the pandas tables using the “seaborn” graphical library. Note that the distinction between overlapping or discrete (non-overlapping) G4 motifs may create issues in the way potential motifs are counted. For this reason, we also provide a % PQS factor, which corresponds to the probability that any given nucleotide in the group or subgroup belongs to a G4-prone region (G4H > 1.2).

The default window value for G4Hunter has been discussed and tested in previous publications [41]. The value is chosen here (25 nt) corresponds more or less to the size of a typical intramolecular quadruplex. We considered shorter windows (20 nt) in previous studies. However, we noticed that for low thresholds (<1.2), a single GGGGGG run would give a hit; while intermolecular G4 formation is indeed possible with this motif, we hypothesized that intramolecular structures would be more relevant.

A slightly longer window (e.g., 30 nucleotides) further contributes to eliminating such motifs, but at the cost of significantly decreasing the number of hits (by a factor of 2 to 3; see Table 1): This larger window would, therefore, increase the number of false negatives, i.e., miss “real” intramolecular G4. On the other hand, a much larger window (50–100 nt) would be interesting to identify “G4 clusters” in which multiple tandem quadruplexes may be formed. We present the number of sequences found in three different complete archaeal genomes using four different window sizes and a threshold of 1.2:

As shown in Table 1, long G-rich prone regions, potentially supporting the formation of multiple quadruplexes, are present, but far less frequent (by a factor of 19 to 186 for a window of 50 vs. 25) than the classically defined G4Hunter motifs. In these three genomes, a large majority (95–99%) of the G4-prone regions would only support the formation of a single individual quadruplex.

### 2.3. Analysis of Putative G4 Sequences Around Annotated NCBI Features

We downloaded feature tables from the NCBI database along with genomic DNA sequences. Feature tables contain annotations of known features found in DNA sequences. We performed an analysis of G4-prone sequences occurrence inside recorded features. Features were grouped by their name stated in the feature table file (gene, rRNA, tRNA, ncRNA, and repeat region). From this analysis, we obtained a file with feature names and numbers of putative G4 forming sequences found inside and around features for each group of species analyzed. Search for putative G4 forming sequences took place inside feature boundaries; note that frequencies of inverted repeats in mitochondrial DNA (mtDNA) [48], as well in the G4 prone sequences in bacteria [33], are distributed with different frequencies in close proximity to specific features. Further processing was performed in Microsoft Excel and the data are available as Appendix A.

### 2.4. Statistical Analysis

A cluster dendrogram of PQS characteristics was constructed in program R, version 3.6.3, library *pvclust* [55], to further reveal and graphically depict similarities between particular archaeal subgroups. Mean, Min, Max, and % PQS values were used as input data (Appendix A). The following parameters were used for analysis: Cluster method ‘ward.D2′, distance ‘Euclidean’, number of bootstrap resampling was set to 10,000. Statistically significant clusters (based on AU values (blue) above 95, equivalent to *p*-values less than 0.05) are highlighted by rectangles marked with broken red lines. R code is provided in Appendix A). Statistical evaluations of differences in G4 forming sequences presence in various phylogenetic groups were made by a Kruskal–Wallis test with a Bonferroni adjustment in STATISTICA, with p-value cut-off 0.05; data are available in Appendix A.

### 2.5. Quadruplex Formation In Vitro

Representative examples of the candidate sequences identified by G4Hunter were experimentally tested for G4 formation using different techniques: Isothermal difference spectra (IDS) and Circular dichroism (CD as described previously [41]).

#### 2.5.1. Samples

Oligonucleotides were purchased from Eurogentec, Belgium, as dried samples purified by RP cartridge purification. Stock solutions were prepared at 250 μM strand concentration in ddH_2_O.

#### 2.5.2. Experimental Conditions

Most experiments were performed in a 10 mM Lithium Cacodylate pH 7.1 buffer supplemented with 100 mM KCl (since *Hadesarchaea* has not been cultivated, it is impossible to know their intracellular potassium concentration. However, this is in the range of intracellular potassium concentration for other archaea, such as *Thermococcales*).

#### 2.5.3. Isothermal Spectra

2.5 µM oligonucleotide solutions were prepared in 10 mM Lithium Cacodylate buffer at pH 7.1. The solutions were kept at 95 °C for 5 min and slowly cooled to room temperature and kept at 4 °C overnight. Absorbance spectra were recorded on a Cary 300 (Agilent Technologies, France) spectrophotometer at 37 °C (scan range: 500–200 nm; scan rate: 600 nm/min; automatic baseline correction). After recording these first series of spectra (unfolded as no potassium was present) 1 M KCl (100 μL) was added to the samples, and UV-absorbance spectra were recorded after 15 min equilibration, and corrected for dilution. Each IDS corresponds to the arithmetic difference between the initial (unfolded) and final (folded, corrected for dilution) spectra.

#### 2.5.4. Circular Dichroism

2.5 µM oligonucleotide solutions were prepared in 10 mM lithium cacodylate buffer at pH 7.1 supplemented with 100 mM KCl. The solutions were kept at 95 °C for 5 min and slowly cooled to room temperature and kept at 4 °C overnight. CD spectra were recorded on a JASCO J-1500 (France) spectropolarimeter at room temperature or at 80 °C, using a scan range of 400–210 nm, a scan rate of 200 nm/min, and averaging four accumulations (Appendix A).

### 2.6. G-Quadruplex Binding Proteins Prediction

For G-quadruplex binding proteins prediction, based on previously published G-quadruplex binding motif (RGRGRGRGGGSGGSGGRGRG) [31], the BLASTp algorithm was used [56]. The target organisms were limited to the Archaea domain (NCBI taxid ID: 2157). E-value cut-off was set to 0.05. For similarity search of RecQ helicase from *Escherichia coli* (UNIPROT ID: P15043), BLASTp algorithm [56] was used with an E-value cut-off of 0.0001 and the same restriction to the Archaea domain, as above. BLASTp analyses are enclosed in Appendix A. FIMO search [57,58] for G-quadruplex binding motif (RGRGRGRGGGSGGSGGRGRG) [31] in *Methanosarcina mazei* complete proteome was carried out on a set of 15722 known protein sequences downloaded from NCBI, with q-value (*p*-value corrected for multiple testing by Benjamini and Hochberg method) cut-off of 0.05 (Appendix A). The most similar protein of RecQ helicase from *Escherichia coli* (UNIPROT ID: P15043) in *Hadesarchaea archaeon* isolate WYZ-LMO6 was searched using tBLASTn [59], and the resulting best hit was translated using Expasy Translate Tool [60,61] and functional domain were visualized using NCBI CDD [62] (Appendix A).

## 3. Results

### 3.1. Prediction of G4 Forming Sequences in Archaea

We analyzed the occurrence of putative G4 sequences (PQS) with G4Hunter in 3387 archaeal genomes. The length of sequenced archaeal genomes in our dataset varied from 100 kbps to 13.4 Mbps (list provided in Appendix A). The average GC content was 46.51%, with a minimum of 24.30% for *Nanobsidianus stetteri* isolate SCGC AB-777 (*Nanoarchaeota*) and a maximum of 70.95% for *Halobacteriales archaeon* SW_7_71_33 (phylum *Euryarchaeota*). Using standard parameters for the G4Hunter search algorithm (window size of 25 and G4HS ≥ 1.2) we found 4,470,813 PQS in these 3387 archaeal genomes using a default threshold of 1.2. The higher the G4HS score is, the higher the stability of the structure. Over 90% and 98% of sequences with a score above 1.2 or 1.5, respectively, were experimentally demonstrated to form a stable quadruplex in vitro [41]. Figure 3A provides an example of G-rich motifs found in archaea with G4HS between 1.32 and 3.0. As expected from previous analyses on eukaryotes and bacteria, most (97%) PQS have a relatively low (1.2 to 1.4) G4Hunter score. More stable motifs are rarer, with a sharp decrease in the number of retrieved sequences with scores above 1.4, as shown in Table 2. Only 132 PQS with a G4Hunter score of 2 or more were found. A summary of all PQS found in ranges of G4Hunter score intervals and precomputed PQS frequencies per 1000 bp is provided in Table 2.

The comparison of G4 prone sequences found in archaea with bacteria genomes revealed that in both domains, frequencies sharply decreased with G4HS as compared to the human genome, in which highly stable G4s are relatively more frequent (see Figure 3B). This result indicates an overall stronger relative selection pressure against stable G4 motifs in both archaea and bacteria as compared to humans, and likely most eukaryotes, as the relative number of G4Hunter high scoring motifs is even higher in yeast [63]. Guo and Bartel suggested that eukaryotes have robust machinery that globally unfolds RNA G-quadruplexes, whereas some bacteria have instead undergone evolutionary depletion of G-quadruplex-forming sequences [64]. Our analysis suggests that archaea behave like bacteria, except for the slight difference found for the most stable motifs (G4HS >2), which were less selected against in archaea than in bacteria.

### 3.2. Variation in Frequency for G4 Forming Sequences in Archaea

The total number of analyzed sequences in particular phylogenetic categories, together with a median length of the genome, shortest genome, longest genome, mean, minimal, and maximal observed frequency PQS per kbp, and total PQS counts are shown in Table 3. For this analysis, Archaea have been divided into five superphyla that form monophyletic assemblages (clades) in the most recent phylogenetic analysis and 41 subgroups that correspond to different taxonomic ranks (suffix *aeota* for phylum, candidate phylum, suffix *ales* for orders). Seven subgroups have an average GC content above 50%, the highest GC content being observed in *Halobacteriales* (63.95%), which is also the archaeal group containing the highest number of available genome sequences–440), all other groups have average GC contents below 50%.

The mean frequency of PQS per kbp for all archaeal genomes was 1.207. The lowest mean frequency was for the *Heimdallarchaeota* (0.273), followed by *Methanococcales* and *Methanobacteriales* (0.39). The highest density of PQS was found in the *Hadesarchaea* subgroup (4.607), followed by *Korarchaeota* (2.626). The highest absolute frequency of PQS was found in *Hadesarchaea archaeon isolate* WYZ-LMO6 with 15.3 PQS per 1000bp (i.e., one quadruplex every 65 bp), and the lowest frequency was found in *Methanobrevibacter* sp. 87.7: Interestingly, only 71 PQS were found in its 1.92 Mb long genome (Appendix A). Detailed statistical characteristics for PQS frequencies per kbp (including mean, variance, outliers) are depicted in boxplots for all inspected subgroups (Figure 4). The *Hadesarchaea* subgroup has a higher PQS frequency in comparison to other subgroups. The comparison of the five main superphyla BAT, Cren, Asgard, Eury, and DPANN (*Diapherotrites*, *Parvarchaeota*, *Aenigmarchaeota*, *Nanoarchaeota*, and *Nanohaloarchaeota*) (Figure 1) revealed the highest mean PQS frequency in Cren superphylum (1.15) and the lowest in Asgard superphylum (0.48). However, the *Hadesarchaea* subgroup, which exhibits the highest frequency among subgroups, is found in the Eury superphylum. The detailed data for superphyla are in Appendix A, for subgroups in Appendix A.

A cluster dendrogram shows the similarities among subgroups based on the PQS data (Figure 5). This dendrogram shows that the *Hadesarchaeota* subgroup is the most distant one (the shortest branch length) compare to other subgroups. The cluster dendrogram based on PQS characteristics is similar to the phylogenetic relationships (see Figure 1). For example, all of the Asgard subgroups (*Odinarchaeota*, *Heimdallarchaeota*, *Thorarchaeota,* and *Lokiarchaeota*) lie close together, in one bigger cluster (Figure 5, left part). Other examples are the *Woesearchaeota*, *Aenigmarchaeota*, and *Nanoarchaeota* subgroups, which are members of the DPANN superphylum, and lie adjacent to each other in PQS based cluster tree. On the other hand, all of the subgroups with the prefix “-thermo”, indicative of high-temperature environments, are clustered together (*Thermoplasmatales*, *Thermococcales*, *Thermoproteales*, and *Geothermarchaeota*). These subgroups are relatively PQS rich, but lack phylogenetical proximity, suggesting that PQS richness does not rely on evolutionary proximity.

We then analyzed the relationship between overall % GC content and PQS frequency (Figure 6). PQS frequencies tend to correlate with GC content as G4-prone motifs need to be relatively G-rich; however, there are interesting exceptions to this rule, and this correlation is poorer than anticipated. Ding et al. already noticed that *Methanomicrobia* and *Thermococci* have greater densities of PQS than the theoretical values based on the GC % of their genomes [35]. Organisms with higher than expected PQS frequencies based on their GC content (over 50% of the maximal observed PQS frequency, Figure 6) are highlighted in color; the whole figure is separated into smaller segments according to inspected G4Hunter score intervals. The most extreme outlier is *Hadesarchaea* archaeon, for which 51% of its genome has a G4Hunter score above 1.2, despite a GC content of 54%, i.e., only modestly above the 46.5% average for all sequences tested here, and far below the most GC rich archaea genomes. Cherry-picked examples of G-rich motifs with high G4 Hunter scores (G4HS) in *Hadesarchaea archaeon* are provided in Table 4. We have also carried out additional statistical evaluation of PQS differences between all groups and subgroups; detailed results are found in Appendix A. Nearly all comparisons were significant, i.e., there are significant differences between PQS frequencies of particular groups and subgroups.

Figure 7 shows the relationship between GC percentage and mean PQS frequencies (or mean percentage of PQS length of the genome) in particular archaeal subgroups. Overall, we found some correlation (although far from perfect, as shown by R^2^ = 0.7) between mean PQS frequencies (expressed as the mean fraction of nucleotides of the genomes involved a PQS motif) and increasing GC % content. The highest mean percentage of PQS length of the genomes was found in subgroup *Hadesarchaea*, in which more than 10% of their genomes are involved in a potential PQS.

### 3.3. Localization of PQS in Genomes

To evaluate the position of PQS in archaeal genomes, we downloaded the described “features” of all archaeal genomes and analyzed the presence of all PQS in annotated sequences (Figure 8). Overall, we find a higher density of G4-prone motifs in non-protein coding RNAs (tRNA, rRNA, and other ncRNA) than in protein-coding genes. G4 density in ncRNA is clearly above average genomic G4 density, while mRNA G4 density is close to the genomic average. This may derive in part from the observation that rRNA and tRNA genes are especially GC-rich in hyperthermophilic archaea, in order to stabilize folding under harsh conditions [65]. On the other hand, we can probably expect a stronger selection pressure against the formation of intramolecular quadruplexes within the relatively small tRNA core, as this would disrupt its three-dimensional shape and alter its biological function. In line with this hypothesis, the PQS frequencies are actually lower in tRNA than in ncRNA and rRNA [66]. Interestingly, the 5′ end of some human tRNA genes is often G-rich and has been reported to allow G4 formation: Ivanov and colleagues have shown that mature cytoplasmic tRNAs are cleaved during stress response to produce tRNA fragments that function to repress translation in vivo and that these bioactive tRNA fragments assemble into intermolecular RNA G4s [67]. The 5′ fragment of tRNA^Ala^ involves a predominant hairpin structure that starts with the 5′-GGGGGU motif, allowing the formation of tetramolecular quadruplex structures with five tetrad layers. Interestingly, tRNA-derived fragments have also been described in archaea. For example, a 26-residue-long fragment (5′ GGGUUGGUGGUCUAGUCUGGUUAUGA) originating from the 5′ part of valine tRNA is the most abundant tRNA fragment in *Haloferax volcanii* [68]. This fragment, while exhibiting a relatively G-rich 5′ end (starting with GGGUUGG), may, in principle, allow intermolecular quadruplex formation as well.

Unfortunately, other features in archaeal genomes are so poorly annotated that we cannot use these data for evaluation. Comparison of PQS frequencies in annotated sequences with analyses of Bacteria shows the same trend for ncRNA, rRNA, protein-coding gene, and tRNA features. In contrast, the frequency in bacteria for ncRNA is 1.7 per kbp, and the frequency in archaea for ncRNA is 5.3 per kbp. On the other hand, the PQS frequency in repeat regions is lower in archaea than in the bacteria genome. We have to take into account that the data could be influenced by poor annotation in archaea genomes, and also by a low number of annotated sequences in Archaea; only 141 representative archaeal genomes are annotated, compared to 1627 representative bacteria annotated genomes. The strong abundance of the PQS in ncRNA compare to other locations pointing to its functional relevance. ncRNAs are present in the cells as single-stranded molecules in contrast to DNA, and therefore, they can easily adopt the G4 structures as a part of their 3D arrangement similarly to mRNAs [69,70]. It has been shown that ncRNAs play important roles in many cellular processes, including the regulation of gene transcription, post-transcriptional, and epigenetic regulations [71,72].

Other specific regions, such as replication origins or promoter regions, were not included in this graph. The oriC 10.0 database (http://tubic.org/doric/public/index.php) contains 226 archaeal origins of replication obtained by both in vitro studies and in silico predictions ([73]), prediction and experimental data are available for the *Thermococcales* [74,75], the *Haloarchaea*, and the *Sulfolobales* [76]. Archaeal replicators, as in bacteria, are composed of three main elements: A cluster of binding sites for the initiator Cdc6, the DNA unwinding element (DUE), and binding sites for regulatory proteins [75]. Interestingly, it was found in several *Haloarchaea* species that a specific (TGGGGGGG) motif occurs in one of the two origins of replication (oriC1) [77]. This long G-rich motif was shown to be necessary for efficient replication initiation in *Haloarcula hispanica* [78,79] and predicted to be prone to inter-molecular quadruplex formation.

### 3.4. Experimental Demonstration of Quadruplex Formation In Vitro

Next, we selected a few DNA G4-prone motifs found in *Hadesarchaea* and experimentally tested if they formed a G4 structure under classical conditions. As inferred from isothermal difference spectra (IDS) (Figure 9a) and circular dichroism (CD) spectra (Figure 9b), all motifs clearly formed G-quadruplexes at room temperature. However, as these motifs are found in an archeon expected to live at a high temperature, we also recorded the spectra at 80 °C. As shown in Figure 9c, these quadruplexes were thermally stable and still formed at high temperatures. Of note, most spectra are indicative of a parallel fold. This bias is the result of a high threshold for G4Hunter (all motifs have scores > 1.7). As a consequence, these motifs are very G-rich, with runs of G separated by short spacers, often 1–2 nt. As short loops tend to be propeller-type, this sequence bias will favor a parallel conformation.

### 3.5. G4-Binding Proteins from Archaea

Given that G4-prone motifs are found in Archaea, and actually extremely abundant in some subgroups, it was interesting to check if potential helicases are present to solve these structures. A number of DNA and RNA G4-helicases have been identified in eukaryotes, e.g., Pif1, DOG, Rhau/DHX36, WRN, BLM; for a review [80]. Little or no experimental data is currently available on archaea enzymes able to unfold G-quadruplexes. As RecQ has been reported to unfold G4 structures in bacteria, we searched for RecQ homologs in Archaea. A BLASTp search using RecQ (UNIPROT ID: P15043) from *E. coli* as a query revealed 1206 homologous protein sequences in a archaeal domain with an E-value cut-off = 0.0001. A listing of all candidates identified is presented in supplementary information (Appendix A). Five proteins have an identity with G-quadruplex RecQ resolvase higher than 50%, and 312 proteins have more than 50% aa positives hits in the sequence, suggesting that they share the G4 unfolding functionality in archaea genomes. Besides protein actively unfolding G4 structures, other peptides may actually bind to single-strand G-rich sequences and passively contribute to G4 unfolding by conformational selection. This is the case for a single-strand binding protein isolated from *Methanococcus jannaschii*, which was used to design an assay to detect G4 formation [79]. Apart from proteins that actively or passively unfold quadruplexes, others may bind to and sometimes promote G4 formation. The amino acid composition of 77 G-quadruplex binding proteins from *Homo sapiens* revealed unique features of quadruplex binding proteins, with prominent enrichment for glycine (G) and arginine I [31]. Human-binding proteins share a 20 amino acid long motif/domain (RGRGR GRGGG SGGSG GRGRG), which is similar to the previously described RG-rich domain of the FMR1 G-quadruplex binding protein. The search for this 20 amino acid-long motif in archaea proteome found 23 hits/potential G-quadruplex binding proteins with an E-value threshold of 0.05; the identity was found, e.g., for RNA DEAD box helicase or for two 30S ribosomal proteins S4 (Appendix A, list 2). We searched protein sequences in the proteome of the mesophylic archaeon *Methanosarcina mazei* (for which the largest amount of proteins is known) for the presence of this motif. For highly significant p values (*p* < 10^−6^), we found four proteins with a potential quadruplex-binding motif (Appendix A), while significantly more (193) hits were found for *p*-values < 1 × 10^−5^. Three of them are without any known function (DUF134 domain-containing protein, PGF-pre-PGF domain-containing protein, and DUF5320 domain-containing protein). Even if the full proteome of *Hadesarchaea archaeon* is not known, it is interesting to note that this RG-domain is present in a number of putative proteins. In addition, while a true RecQ homolog was not found, one *Hadesarchaea archaon* 600aa-polypeptide has a good similarity with RecQ in its N-terminal half (Appendix A). The presence of the NIQI motif in the “DNA-directed RNA polymerase subunit” is also interesting and possibly logical, given the necessity of unraveling G-quadruplexes during transcription. The presence in archaeal genomes of potential G4-binding and G4-unfolding proteins supports the formation of quadruplex structures in archaeal cells.

## 4. Discussion

We provide here the first comprehensive study of PQS occurrences, frequencies, and distributions in archaeal genomes. The overall analysis made on global frequency hides extreme differences between species and subgroups, which can be explained by differences in GC content and possibly codon usage.

At one end of the G4 spectrum, some subgroups of archaea, such as *Parvarchaeota* or *Heimdallarchaeota*, have very low PQS frequencies, and PQS cover 1% or less of their genomes. In sharp contrast, we found an unprecedented enrichment of PQS for some subgroups, often living under extreme conditions. For example, over 50% of the genome of *Hadesarchaea archaeon* may potentially adopt a quadruplex fold. This *Hadesarchaea* is living under extreme conditions, as it was found in South African gold mines 3 km underground, without light and oxygen (*Hades* is the Greek god of the underworld). Following this analysis, we used the BioSample NCBI database [78] to compare the living environment of the archaea organisms with the highest PQS frequencies. Data for all genomes with PQS frequency above 6 per kbp are shown in Table 5. A majority of organisms with extremely high PQS frequencies are found in hot springs sediments or in deep-sea hydrothermal vent sediments, and this high PQS frequency may be associated with their extremophilic life, although more work will be necessary to compare G4 density in acidophilic, thermophilic, halophilic and psychrophilic organisms. For example, in bacteria, in the Gram-positive subgroup *Deinococcus-Thermus*, a high PQS frequency was associated with their extremophilic origin [35,81], while the gram-negative extremophilic bacteria subgroup *Thermotogae* are among organisms with a low PQS frequency [33]. We suggest that the high stability of G4 structures compare to dsDNA structure could play important roles in archaea and Gram-positive extremophiles organisms. We then experimentally confirmed G4 formation with a few archaea sequences to confirm that our in silico predictions are verified: All predicted experimentally tested formed stable G-quadruplexes in vitro. This absence of false positives is hardly surprising given that we chose high scoring motifs. From our published [41] and unpublished data on now over 500 sequences, false positives for sequences with scores above 1.5 are extremely rare (<1.5%), and we have yet to find a false positive with a score > 1.75. Some of the sequences considered were long and may even allow the formation of two juxtaposed G4 structures. In a few cases, we can even propose a topology, as for example, TGGTGGGGGCGGGGGGAGGGGCGGGGGT (642K), in which the predicted guanine tracks (underlined) may either be: TGGTG**GGGG**CG**GGGG**GA**GGGG**CG**GGGG**T or TGGTG**GGGG**C**GGGG**GGA**GGGG**C**GGGG**GT, and different folds may result from these possibilities (the latter would be likely parallel, as experimentally observed at 80 °C, while the former may adopt a non-parallel fold, as observed at room temperature). Note, however, that G4 hunter does not make any hypothesis on the G tracts involved in G4 formation, in contrast with Quadparser, for example, where one actively seeks the four runs of G involved in G-quartet formation. G4 formation is (still) full of surprises, and correctly predicting which runs (or individual guanines) participate in G-quartet formation is far from trivial and requires extensive experimental validation.

The extreme enrichment found in some archaea challenges our existing views on “noncanonical” DNA structures to which G-quadruplexes belong, as it is plausible that a substantial part of the *Hadesarchaea* genome may be packed into G-quadruplex structures. The complementary C-rich strand may also fold into a different quadruplex structure called the i-motif [82] that is favored by acidic pH. Further studies will be dedicated to i-DNA formation in Archaea.

*Hadesarchaea archaeon* isolates WYZ-LMO4, WYZ-LMO5, WYZ-LMO6 are archaeal species isolated from hydrothermal spring sediments. Besides high temperatures, often above 50 °C, these ecological niches usually have high salinity. Interestingly, most G-quadruplexes withstand high temperatures (their melting point is often above 70 °C) and are further stabilized by positively charged ions such a K^+^ and Na^+^ [84,85]. Such conditions may have naturally favored G-quadruplexes over duplexes. It also highlights one of the consequences of a high GC %: G4-prone motifs become more frequent (Figure 5). In addition, all hyperthermophilic organism genomes encode a reverse gyrase, which positively supercoil DNA, possibly to protect the genome [86]. In future studies, it would be very interesting to carry out a genome-wide wet-lab experiment, for example, direct DNA sequencing of G-quadruplex loci as described in [87,88] or direct visualization of G-quadruplexes in living cells using specific antibodies, such as BG4 [89].

## 5. Conclusions

Overall, our results indicate that archaea are, like eukaryotes and bacteria, prone to G-quadruplex formation: G-quadruplexes are here, there, and everywhere! Important differences in G4 densities were found among species, and experimental validation was obtained in vitro for a few candidate sequences. Follow-up studies may check if specific archaeal PQS loci—for example, in important genes, show some phylogenetic conservation. If confirmed, this could serve as a new (additional) phylogenetic marker and give us some extended clues about the evolution and function of G-quadruplex forming sequences in Archaea. This study will stimulate further studies on G4 presence in Archaea, and help to establish whether some regulatory mechanisms may only apply to a given domain or be truly universal.

## Figures and Tables

**Figure 1 biomolecules-10-01349-f001:**
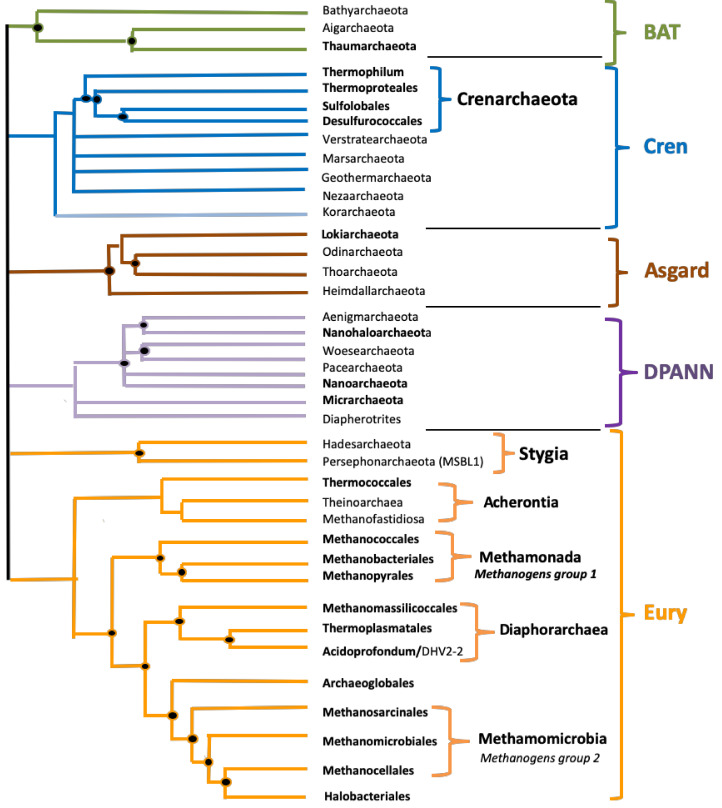
A schematic phylogenic tree for Archaea. This unrooted evolutionary tree of Archaea is based on the schematic tree of Forterre (2015) [17] updated according to recent phylogenetic analyses [9,18]. BAT stands for Bathyarchaeota, Aigarchaeota, and Thaumarchaeota. DPANN is an acronym based on the first five groups discovered: *Diapherotrites*, *Parvarchaeota*, *Aenigmarchaeota*, *Nanoarchaeota*, and *Nanohaloarchaeota*. The term BAT superphylum has been proposed by Gaia et al. in 2018 [19], and the terms Eury and Cren superphyla are suggested here. The terms Cren superphylum is suggested here because the phyla *Crenarchaeota*, *Verstratearchaeota Marsarchaeota*, *Nezaarchaeota*, and Geothermarchaeota form a consensus monophyletic clade in all archaeal phylogeny. We included *Korarchaeota* in this superphylum because they often branch as sister groups of the above phyla in archaeal phylogenies, although the fast evolutionary rate made their positioning sometimes difficult. We suggested in parallel the term Eury superphylum because Euryarchaeota includes very diverse groups of cultivated and uncultivated Archaea which are difficult to the group in a single phylum, especially considering that phyla, such as *Verstratearchaeota Marsarchaeota*, or *Nezaarchaeota* only contain few uncultivated species only defined by a few metagenome associated genomes (MAGs). Names in bold letters correspond to subgroups that include cultivated species; names in thin letters correspond to subgroups that include only MAGs.

**Figure 2 biomolecules-10-01349-f002:**
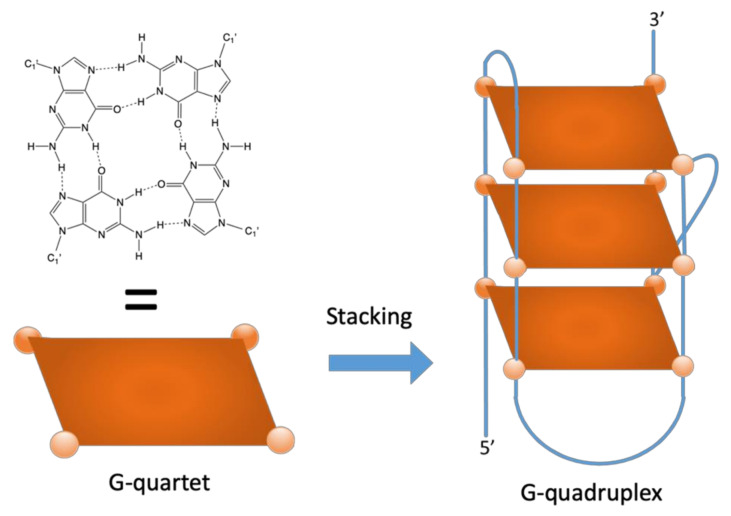
A G-quartet involves four coplanar guanines establishing a cyclic array of H-bonds (left). Stacking of two or more (three in this example) quartets leads to the formation of a G-quadruplex structure (right), stabilized by cations, such as potassium (not shown).

**Figure 3 biomolecules-10-01349-f003:**
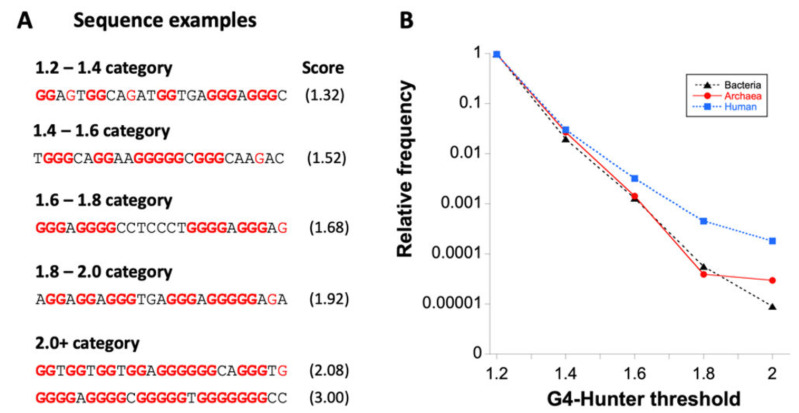
Examples of sequences with different G-quadruplexes (G4) Hunter scores (G4HS) and distribution of PQS according to threshold category. (**A**) Examples of archaea 25-nt long sequences (corresponding to the window size chosen for the analysis) for which G4Hunter scores are provided within parentheses. Isolated guanines are shown in red, all other guanines in bold red characters. Longer archaea motifs with high G4H scores are provided in Table 3. (**B**) Distribution of G4-prone motifs according to the G4Hunter score. 1.2 means any sequence with a score between 1.2 and 1.399; 1.4 between 1.4 and 1.599, etc. These numbers are normalized by the total number of PQS found in bacteria, archaea, and compared with *Homo sapiens*. The first category represents 97.9% and 97.2% of all PQS sequences in bacteria and archaea, respectively. Note the log scale on the Y-axis.

**Figure 4 biomolecules-10-01349-f004:**
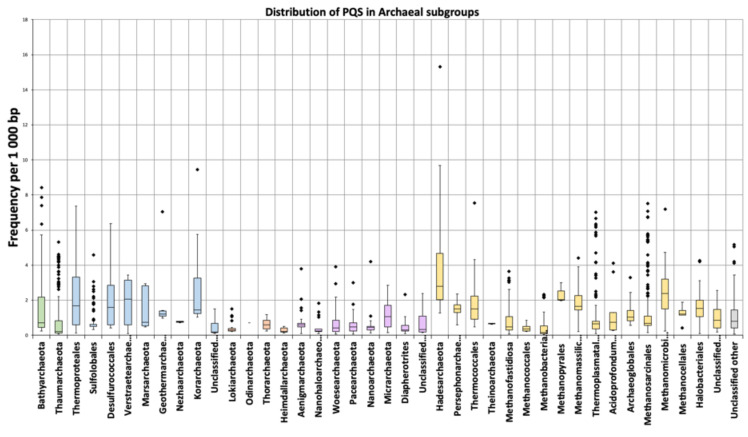
Frequencies of PQS in subgroups of analyzed archaeal genomes. Data within boxes span the interquartile range, and whiskers show the lowest and highest values within 1.5 interquartile range. Black points denote outliers. Horizontal black lines inside boxplots are median values.

**Figure 5 biomolecules-10-01349-f005:**
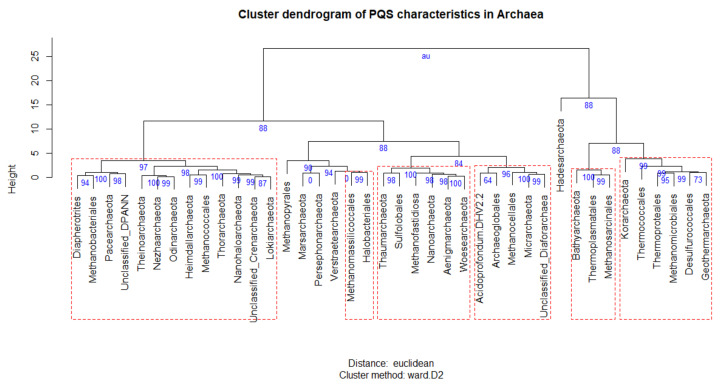
Cluster dendrogram of PQS characteristics of archaeal subgroups. Cluster dendrogram of PQS characteristics (Appendix A) was made in R v. 3.6.3 (code provided in Appendix A) using pvclust package with these parameters: Cluster method ‘ward.D2′, distance ‘euclidean’, number of bootstrap resamplings was 10,000. AU values are in blue and indicate the statistical significance of particular branching (values above 95 are equivalent to *p*-values lesser than 0.05). Statistically significant clusters are highlighted by red dashed rectangles.

**Figure 6 biomolecules-10-01349-f006:**
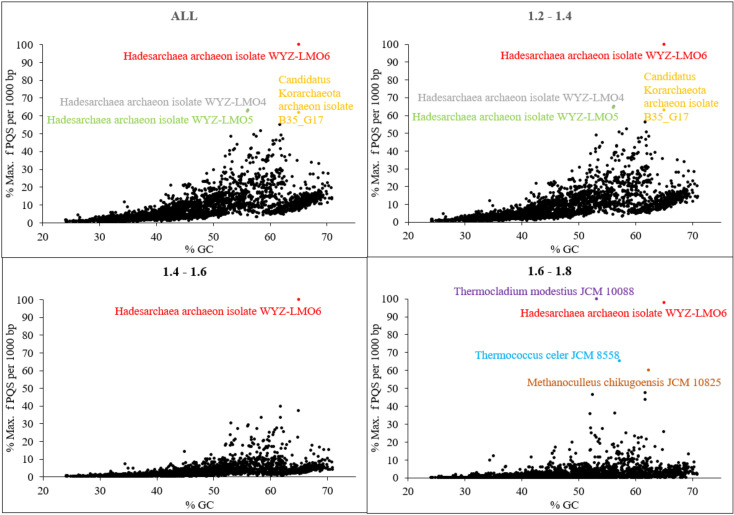
Relationship between the observed frequency of PQS per 1000 bp and GC content. Different G4Hunter score intervals are considered. In each G4Hunter score interval miniplot, frequencies were normalized according to the highest observed frequency of PQS. Organisms with max. frequency per 1000 bp greater than 50% are described and highlighted in color.

**Figure 7 biomolecules-10-01349-f007:**
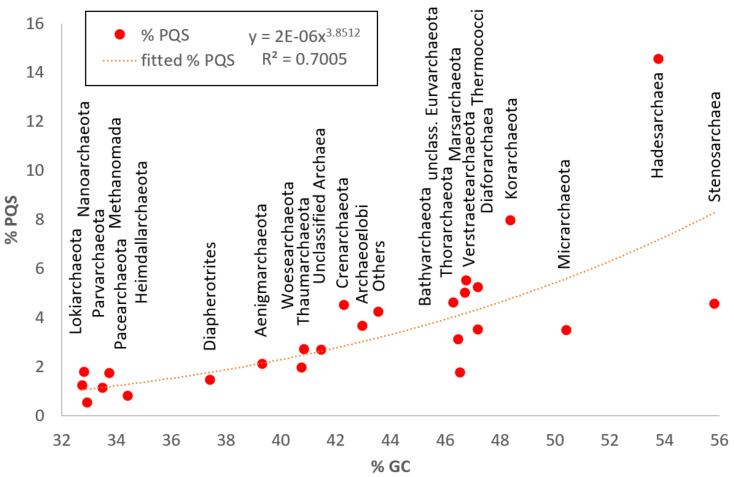
Relationship between GC percentage and % of PQS in genomes of particular archaeal subgroups. The Fitted equation with the R^2^ coefficient is depicted on the top side of the plot.

**Figure 8 biomolecules-10-01349-f008:**
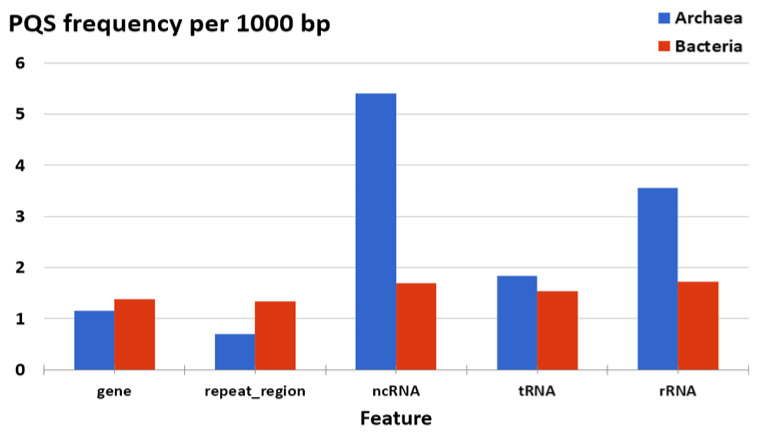
Differences in PQS frequency by DNA locus. The chart shows PQS frequencies normalized per 1000 bp annotated locations from the NCBI database and shows a comparison between Archaea and Bacteria. Archaea G4-prone motifs are strongly over-represented in ncRNA and rRNA compared to the average G4 density in Archaea (mean f = 1.207), but also compared to bacteria. PQS count is provided in Appendix A Excel file.

**Figure 9 biomolecules-10-01349-f009:**
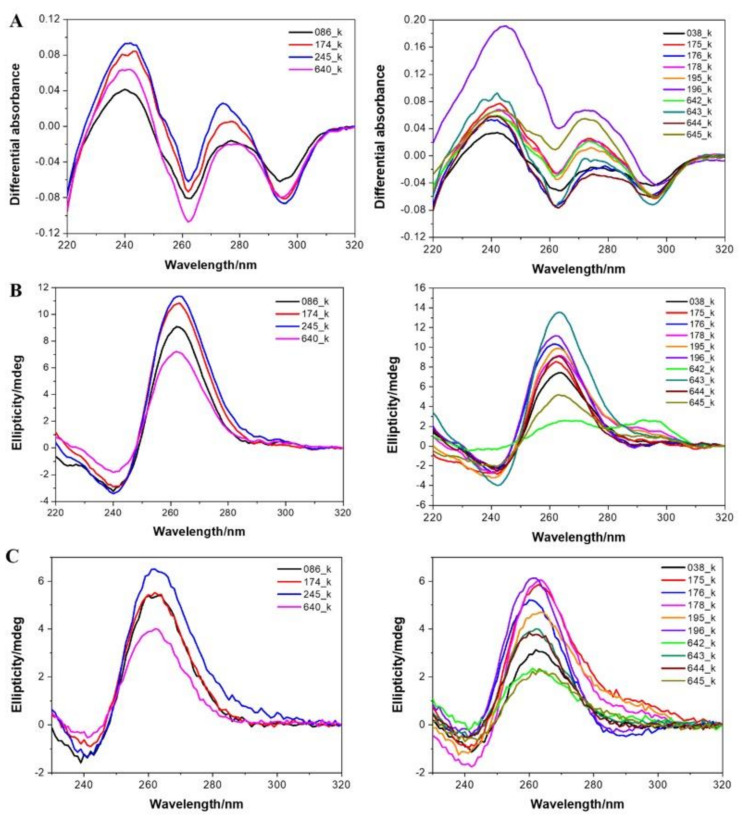
Experimental evidence for quadruplex formation with archaea sequences. Isothermal differential absorbance (IDS; panel **A**) and circular dichroism (CD; panels **B** and **C**) spectra of *Hadesarchaea archeon* DNA sequences were recorded at 20 °C (panels **A** and **B**) or at a high temperature (80 °C) for CD (panel **C**).

**Table 1 biomolecules-10-01349-t001:** A number of putative quadruplex sequences (PQS) were found using four different window sizes in three complete archaeal genomes.

Archaea (GC %)	Number of G4 Sequences Found for a Window of:			
	25 nt	30 nt	50 nt	100 nt
*Methanococcus maripaludis* C7 (33.3%)	558	171	3	0
*Cenarchaeum symbiosum* A (57.3%)	6019	3197	324	5
*Halobacterium salinarum* NRC (65.9%)	4738	2313	262	4

**Table 2 biomolecules-10-01349-t002:** Number of PQS found and their frequencies per 1000 bp in all 3387 archaeal genomes, grouped by G4Hunter score (1.2-1.4 means any sequence with a score between 1.2 and 1.399; 1.4 between 1.4 and 1.599, etc.).

G4HS	Number of PQS in Dataset	Fraction of All PQS	PQS Frequency Per kbp
1.2–1.4	4,344,917	0.9718	1.19
1.4–1.6	119,233	0.0267	1.8 × 10^−2^
1.6–1.8	6357	0.00142	9.9 × 10^−4^
1.8–2.0	174	0.0000389	2.5 × 10^−5^
>2.0	132	0.0000295	2.2 × 10^−5^
Total	4,470,813	1	

**Table 3 biomolecules-10-01349-t003:** Genomic sequences sizes, GC%, total count of PQS, and mean frequencies of quadruplex motifs. Seq (total number of sequences), Median (median length of sequences), Short. (shortest sequence), Long. (longest sequence), GC % (average GC content), PQS (total number of predicted PQS), Mean f (mean frequency of predicted PQS per 1000 bp), Min f (lowest frequency of predicted PQS per 1000 bp), Max f (highest frequency of predicted PQS per 1000 bp). %PQS corresponds to the probability that any given nucleotide in the group or subgroup belongs to a G4-prone region (G4H > 1.2). Colors correspond to phylogenetic tree depiction.

Kingdom	Seq.	Median	Short	Long	GC %	PQS	Mean f	Min f	Max f	% PQS
Archeae	3387	1,686,930	100,212	13,399,915	46.51	7,927,775	1.21	0.04	15.31	3.58
**Superphylum**	**Seq.**	**Median**	**Short**	**Long**	**GC %**	**PQS**	**Mean f**	**Min f**	**Max f**	**% PQS**
BAT	320	1,180,629	164,795	3,506,105	43.07	421,678	1.16	0.05	8.42	3.49
Cren	379	1,808,184	210,860	6,451,204	43.05	1,009,660	1.56	0.09	9.44	4.75
Asgard	71	2,322,715	291,515	5,684,038	38.75	74,647	0.47	0.12	1.50	1.39
DPANN	309	832,169	100,212	6,604,953	39.22	219,058	0.70	0.08	4.20	2.18
Eury	2308	1,826,841	137,797	13,399,915	48.77	6,202,732	1.25	0.04	15.31	3.68
**Phylum**	**Seq.**	**Median**	**Short**	**Long**	**GC %**	**PQS**	**Mean f**	**Min f**	**Max f**	**% PQS**
Bathyarchaeota	128	1,208,976.5	200,493	3,506,105	46.29	245,162	1.54	0.23	8.42	3.00
Thaumarchaeota	192	1,173,909.5	164,795	3,441,569	40.93	176,516	0.91	0.05	5.32	2.73
Thermoproteales	147	1,581,744	242,587	3,969,448	45.86	513,053	2.07	0.11	7.38	6.31
Sulfolobales	118	2,223,757.5	210,860	3,034,024	38.20	200,842	0.79	0.34	4.58	2.38
Desulfurococcales	29	1,580,347	807,477	2,148,448	46.99	99,211	2.29	0.40	6.37	6.95
Verstraetearchaeota	18	1,171,913.5	419,172	1,937,662	46.76	40,586	1.83	0.10	3.43	5.50
Marsarchaeota	15	1,915,630	351,358	3,731,392	46.72	52,853	1.64	0.47	2.94	5.01
Geothermarchaeota	6	1,183,145.5	803,797	1,671,866	42.72	16,582	2.15	0.96	7.03	6.65
Nezhaarchaeota	2	1,332,140.5	1,315,707	1,348,574	43.53	2016	0.76	0.75	0.77	2.27
Korarchaeota	18	1,542,873	834,209	2,942,065	48.39	68,434	2.63	1.05	9.44	7.95
Unclassified Crenarchaeota	27	1,203,892	301,027	6,451,204	37.01	19,361	0.44	0.09	1.49	1.29
Lokiarchaeota	29	1,892,624	320,847	5,143,417	32.77	25,479	0.41	0.21	1.50	1.24
Odinarchaeota	1	1,460,710	1,460,710	1,460,710	38.05	1038	0.71	0.71	0.71	2.16
Thorarchaeota	29	2,770,204	291,515	4,389,059	46.55	40,006	0.60	0.24	1.18	1.76
Heimdallarchaeota	12	2,167,091	432,340	5,684,038	34.42	8124	0.27	0.12	0.50	0.82
Aenigmarchaeota	35	751,672	248,182	1,410,470	39.33	17,990	0.71	0.11	3.78	2.12
Nanohaloarchaeota	17	815,638	565,289	1,480,846	44.53	8672	0.48	0.09	1.82	1.50
Woesearchaeota	72	966,794.5	518,295	2,944,567	40.77	57,833	0.66	0.08	3.92	1.96
Pacearchaeota	60	719,507	279,432	6,604,953	33.74	37,675	0.56	0.08	2.99	1.73
Nanoarchaeota	25	577,110	204,081	1,162,239	32.83	9940	0.59	0.13	4.20	1.70
Micrarchaeota	39	887,931	658,716	1,333,875	50.41	42,298	1.17	0.15	2.86	3.47
Diapherotrites	19	568,419	302,064	1,130,899	37.42	6077	0.49	0.11	2.33	1.46
Unclassified DPANN	40	858,043.5	100,212	3,188,023	35.57	33,846	0.67	0.15	2.39	2.04
Hadesarchaeota	12	857,575	451,393	1,241,441	53.77	56,369	4.61	1.26	15.31	14.55
Persephonarchaeota	33	637,942	137,797	1,412,535	44.06	34,905	1.49	0.59	2.36	4.49
Thermococcales	60	1,867,904.5	207,909	2,388,527	46.77	191,492	1.72	0.47	7.53	5.15
Theinoarchaeota	2	4,165,806	3,559,548	4,772,064	41.57	5480	0.66	0.65	0.67	1.94
Methanofastidiosa	96	992,372	156,656	13,399,915	40.71	141,192	0.83	0.08	3.64	2.54
Methanococcales	24	1,717,483	1,207,361	1,936,387	32.01	15,065	0.39	0.20	0.86	1.19
Methanobacteriales	224	2,001,036	1,157,521	3,466,370	33.62	175,191	0.39	0.04	2.32	1.14
Methanopyrales	3	1,430,309	1,421,621	1,694,969	58.94	10,798	2.34	1.97	3.00	6.84
Methanomassilicoccales	91	1,404,109	640,223	2,641,216	56.22	257,340	1.85	0.22	4.41	5.38
Thermoplasmatales	135	1,621,237	593,453	2,816,557	42.71	246,832	1.13	0.11	7.03	3.42
Acidoprofondum/DHV2-2	11	1,731,076	519,420	2,981,805	40.55	16,609	1.21	0.29	4.12	3.59
Archaeoglobales	53	1,901,943	478,535	3,408,041	42.98	117,470	1.22	0.57	3.29	3.66
Methanosarcinales	279	2,913,215	208,261	5,751,492	44.99	845,394	1.19	0.15	7.52	3.54
Methanomicrobiales	146	2,228,967.5	622,799	3,978,804	54.97	783,172	2.38	0.23	7.20	7.07
Methanocellales	5	2,957,635	1,465,272	3,243,770	50.96	16,825	1.21	0.41	1.88	3.51
Halobacteriales	440	3,585,981	397,623	5,605,381	63.95	2,271,600	1.56	0.08	4.25	4.50
Unclassified Diaforarchaea	97	1,460,542	233,168	2,294,894	47.38	136,115	1.03	0.18	2.55	3.02
Unclassified other	597	1,400,198	258,312	7,416,915	46.88	862,962	1.02	0.07	5.16	3.00

**Table 4 biomolecules-10-01349-t004:** Long G4-prone motifs with high G4HS found in *Hadesarchea archeon*.

Name	Sequences (5′ to 3′)	G4 Hunter Score	IDS	CD
038_K	AGGCTGGGGGTGAGGGCGGTGGTGGGGAAGGGAGGGGTGGGGGAGAAAACGAAGGGGGT	2.07	G4	Parallel
086_K	TGGGGAGGAGGGGAGGGGAGGTGGGCTGGGGGGGGCT	2.57	G4	Parallel
174_K	AGGGTGAGGGAGGAGGTGCTGGGGGGAAGGGAGGTGGGGGAGGGGGAGGTGGAGGGGCTGGTGAGGGA	2.07	G4	Parallel
175_K	AGGGGAGGAGGGTGGCCGTGGTGGGGGCGGGGGGAGGGGCGGGGGTGGGGGGGCCTGGGGGGA	2.54	G4	Parallel
176_K	AGGAGGAGGGTGAGGGACCAGGGGAGGAGGGAGGGGAGGGGGGGAAGGAGGAGGGAGAGGAGGAGGGA	1.93	G4	Parallel
178_K	TGGTGGGGGCGGGGGGAGGGGCGGGGGTGGGGGGGCCTGGGGGGA	2.89	G4	Parallel
195_K	AGGGGAGGAGGGTGGCCGTGGTGGGGGCGGGGGGAGGGGCGGGGGTGGCCTCCACGGA	1.91	G4	Parallel
196_K	AGGGGAGGAGGGAGGGGAGGGGGGGAAGGAGGAGGGAGAGGAGGAGGGA	2.22	G4	Parallel
245_K	GGGGTCGTCGGGGGGGAGAGCTGGGGAGGAGGGGAGGGGAGGTGGGCTGGGGGGGGCTGGGGAGGGAGGAGGTGAGGGG	2.33	G4	Parallel
640_K	AGGGAGGTGGGGGAGGGGGAGGTGGAGGGGCT	2.38	G4	Parallel
642_K	TGGTGGGGGCGGGGGGAGGGGCGGGGGT	2.93	G4	Hybrid*
643_K	AGGCTGGGGGTGAGGGCGGTGGTGGGGAAGGGAGGGGTGGGGGAGAAAACGAAGGGGGT	2.07	G4	Parallel
644_K	AGGGCGGTGGTGGGGAAGGGAGGGGTGGGGGA	2.41	G4	Parallel
645_K	GGCGGGGGGGGAGTCCTTCATCCTGGGGTAGGGG	1.74	G4	Parallel

* Sequence 642_K adopts a hybrid structure at room temperature, which is converted to a parallel conformation at high temperatures.

**Table 5 biomolecules-10-01349-t005:** Detailed characteristics of archaeal species with PQS frequency per 1000 bp greater than 6.00. Living environments data were obtained from the BioSample NCBI database [83].

Organism Name	GC Content	PQS f	% PQS	Living Environment (Isolated from)
*Hadesarchaea archaeon* isolate WYZ-LMO6	65.01	15.310	51.15	Hot springs sediment, Yellowstone NP, USA
*Hadesarchaea archaeon* isolate WYZ-LMO4	56.17	9.685	31.10	Hot springs sediment, Jinze hot spring, China
*Hadesarchaea archaeon* isolate WYZ-LMO5	56.04	9.581	30.69	Hot springs sediment, Jinze hot spring, China
*Korarchaeota archaeon* isolate B35_G17	65.01	9.445	28.80	Deep-sea hydrothermal vent sediments, Guaymas Basin, Gulf of California, Mexico
*Bathyarchaeota archaeon* B23	61.78	8.418	26.12	Deep-sea hydrothermal vent sediments, Guaymas Basin, Gulf of California, Mexico
*Bathyarchaeota archaeon* isolate M10_bin139	58.42	7.858	24.55	Deep-sea hydrothermal vent sediments, Guaymas Basin, Gulf of California, Mexico
*Thermococcus celer* JCM 8558	57.21	7.534	24.52	Solfataric marine water hole on a beach of Vulcano, Italy
*Methanosaeta harundinacea* isolate UBA152	62.01	7.518	23.12	Waste water, Suncor tailings pond 6, Canada
*Bathyarchaeota archaeon* isolate B23_G15	57.67	7.397	22.90	Deep-sea hydrothermal vent sediments, Guaymas Basin, Gulf of California, Mexico
*Thermocladium modestius* JCM 10088	53.14	7.381	25.59	Mud from a spring pool, Noji-onsen, Fukushima, Japan
*Methanoculleus chikugoensis* JCM 10825	62.36	7.198	22.90	Paddy field soil, Chikugo, Fukuoka, Japan
*Methanosaeta harundinacea* isolate UBA281	61.14	7.089	21.80	Wastewater, North Alberta, Canada
*Geothermarchaeota* archaeon ex4572_27	60.54	7.032	22.01	Deep-sea hydrothermal vent sediments, Guaymas Basin, Gulf of California, Mexico
*Thermoplasmata archaeon* isolate CSSed11_322R1	61.82	7.028	22.57	Hypersaline soda lake sediment, Kulunda Steppe, Russia
*Methanosarcinales* archaeon Methan_02	60.8	6.738	20.67	Anaerobic digester metagenome, Australia
*Methanosaeta harundinacea* 6Ac	60.6	6.721	20.66	isolated from an upflow anaerobic sludge blanket reactor treating beer-manufacture wastewater in Beijing, China. (ref PMID:16403877)
*Thermoplasmatales* archaeon ex4484_36	54.25	6.673	21.15	Deep-sea hydrothermal vent sediments, Guaymas Basin, Gulf of California, Mexico
*Aeropyrum camini* SY1 = JCM 12091	56.73	6.370	19.72	Deep-sea hydrothermal vent chimney, the Suiyo Seamount in the Izu-Bonin Arc, Japan
*Bathyarchaeota archaeon* isolate B46_G17	61.92	6.332	19.03	Deep-sea hydrothermal vent sediments, Guaymas Basin, Gulf of California, Mexico
*Thermoplasmata* archaeon isolate B14_G15	53.83	6.327	20.11	Deep-sea hydrothermal vent sediments, Guaymas Basin, Gulf of California, Mexico
*Thermoplasmata* archaeon isolate B23_G1	53.66	6.240	19.72	Deep-sea hydrothermal vent sediments, Guaymas Basin, Gulf of California, Mexico
*Pyrobaculum neutrophilum* V24Sta	59.91	6.233	19.52	isolated from a hot spring in Iceland
*Thermoplasmata* archaeon isolate B23_G9	52.98	6.164	19.65	Deep-sea hydrothermal vent sediments, Guaymas Basin, Gulf of California, Mexico

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
