# Peer review of "G-Quadruplexes in the Archaea Domain"

_biomolecules, 2020, doi:10.3390/biom10091349_

Round 1

Reviewer 1 Report

The manuscript deals with the identification of putative G-quadruplexes (PQS) in the Archaea domain using the G4Hunter tool.

The whole manuscript is well written and despite mainly an in-depth in silico analysis (though limited by the use of G4Hunter only), there are also some partial confirmations of the results by Circular dichroism (CD).

There are still, a few major points that the authors are requested to address:

1)      G4Hunter has been used with default parameters of window size 25nt. Is there any particular reason? Do you have tested other window ranges? The use of different window sizes may affect the final score for the detected regions and the authors are requested to prove and test other window sizes to be compared on presented data

2)      Table 2 shows the results of highly scored regions detected by G4Hunter. It is not completely clear the choice of these sequences. Are they the highest top-ranked regions? Are they a selection of some kind (cherry-picking)? Have you tested other regions that proved not to form G4s in vitro (i.e. G4Hunter false positives)? Reporting potential negative results would be of great interest and importance

3)      What is helpful in the study of a particular region (as those shown in Table 2) containing a PQS, is to have an idea of what are the G-islands that are putatively involved in the G4 formation. Looking at the sequences and taking as an example 245_K, it is very long with a strong CD signal but there are plenty of combinations that may give a G4, and possibly this region could form even two close G4s. On the contrary, 645_K (a not very “important” CD signal according to supplementary material) is very short and probably with few alternatives. I ask the authors to give their predictions of the potential islands involved in the formation of a G4 with the help of different tools specialized in this task as recent QPARSE (Berselli et al. Bioinformatics 2020) and/or PQSFINDER (Labudova et al. Bioinformatics 2020). This could help readers that are interested in the details of a region, not to be puzzled by the manifold combinations that a G-rich region may offer. Hence, proposing a starting point with some reasonable hypotheses is necessary.

Minor issues (but not less important)

1)      The use of homology must be avoided in some of the contexts where it is used in the manuscript. Line 190 is one of them, for example. BLAST can perform similarity searches and not homology searches that are a completely different thing. I suggest the reading of Walter Fitch (Trends In Genetics May 2000, volume 16, No. 5) so you will realize that similarity and homology are non-synonyms. Some of the searches that you perform and hits you get with a low e-value can be also due to analogy. You can assert either of the two only after a careful phylogenetic analysis. Even homologic in supplementary material should be revisited unless a clear evolutionary origin (homology) is ascertained using phylogenetic analysis tools.

2)      Some scattered typos to check. 

Reviewer 2 Report

The manuscript by Václav Brázda et al. reports the first bioinformatics analysis of the occurrence of G-quadruplexes (G4s) in the genomes of the archaea domain species. Furthermore, selected G-rich sequences were shown to fold into stable G4s in vitro using circular dichroism. Overall, the authors present a sound study on the occurrence and ubiquity of G4 regulatory regions across species other than bacteria and eukaryotes. Therefore, the topic is of interest for the Biomolecules journal readership and is recommended for publishing after the following major and minor issues have been addressed:

Major points

  • What is the rationale behind Eury and Cren superphyla terms?
  • The archaea have been described for colonizing the human microbiota. Is there any possible interaction with the host regulated by G4-driven mechanisms?
  • Figure 3B: Is there any possible comparison (e. similar genomic location, function, …) between the found PQS with a G4Hunter score of 2 or more between human, bacteria and archea?
  • Are there any possible relationships between the formation of G4s in the genome of archaea and their resistance to harsh environments? For instance, are genes regulated by these harsh environments enriched with G4s? This is particularly interesting as extremophile bacteria are also known for being G4-rich.
  • Sequence 642_K is listed in Table 2 as a hybrid-type G4. However, in Figure S1 the CD spectra suggests a parallel topology. Please explain.
  • Are the tested potassium conditions (100 mM KCl) physiologically relevant for Hadesarchaea?
  • All the tested DNA sequences seem to fold into a parallel G4 topology which is common for RNA sequences but not for DNA. Is this related to their stability? Please comment.

Minor points

  • Please indicate the degree of purification of the purchased oligos.
  • Please indicate the working concentration of the oligos and salts in the biophysical experiments, rather than the added volumes.
  • The authors state that “the highest GC content being observed in Halobacteriales (65.95%)” however in Figure 4 the percentage is lower, 63.95%.
  • Figure 4 should be Table 4.
  • Figure 7 captions need reformatting.
  • Table 2: what does the asterisk mean?
  • Please provide the CD spectra for all DNA sequences at room temperature (rather than just at 80 °C like Fig. S1B), at least as supporting figure.

Round 2

Reviewer 1 Report

The authors have partially answered the requests. Still missing the prediction of PQS (as an indication of what are the potential combinations of G-islands involved) from the detected G-rich regions showing to fold in G4. I respectfully disagree that authors want to refrain to propose some in silico solutions, but at a second evaluation, I accept their point of view in the end.